# What Psychosocial and Physical Characteristics Differentiate Office Workers Who Develop Standing-Induced Low Back Pain? A Cross-Sectional Study

**DOI:** 10.3390/ijerph17197104

**Published:** 2020-09-28

**Authors:** Beatriz Rodríguez-Romero, Michelle D Smith, Alejandro Quintela-del-Rio, Venerina Johnston

**Affiliations:** 1Psychosocial Intervention and Functional Rehabilitation Research Group, Department of Physiotherapy, Medicine and Biomedical Sciences, Faculty of Physiotherapy, Universidade da Coruña, Campus de A Coruña, 15071 A Coruna, Spain; 2School of Health and Rehabilitation Sciences, The University of Queensland, Brisbane 4072, Australia; m.smith5@uq.edu.au (M.D.S.); v.johnston@uq.edu.au (V.J.); 3Psychosocial Intervention and Functional Rehabilitation Research Group, Department of Mathematics, Faculty of Physiotherapy, Universidade de A Coruna, 15071 A Coruna, Spain; alejandro.quintela@udc.es; 4Recover Injury Research Centre, The University of Queensland, Brisbane 4006, Australia

**Keywords:** low back pain (LBP), standing position, musculoskeletal pain, sedentary behaviour

## Abstract

This study examines demographic, physical and psychosocial factors associated with an increase in low back pain (LBP) during a one-hour standing task. A cross-sectional survey with 40 office workers was conducted. The primary outcome was pain severity during a one-hour standing task recorded every 15 min using a 100 mm Visual Analogue Scale (VAS). Participants were defined as pain developers (PD), if they reported a change in pain of ≥10 mm from baseline, or non-pain developers (NPD). Physical outcomes included participant-rated and examiner-rated trunk and hip motor control and endurance. Self-report history of LBP, physical activity, psychosocial job characteristics, general health and pain catastrophising were collected. Fourteen participants were PD. Hip abduction, abdominal and spinal muscle endurance was lower for PD (*p* ≤ 0.05). PD had greater self-reported difficulty performing active hip abduction and active straight leg raise tests (*p* ≤ 0.04). Those reporting a lifetime, 12 month or 7-day history of LBP (*p* < 0.05) and lower self-reported physical function (*p* = 0.01) were more likely to develop LBP during the standing task. In conclusion, a history of LBP, reduced trunk and hip muscle endurance and deficits in lumbopelvic/hip motor control may be important to consider in office workers experiencing standing-induced LBP.

## 1. Introduction

Low back pain (LBP) is recognised as a serious health concern and is the leading cause of activity limitation and work absenteeism internationally [1]. It is a multi-factorial and heterogeneous condition including complex interactions between physical, psychological, social and comorbid health factors that is best explained by a biopsychosocial framework [2].

Regarding workplace factors and the occurrence of LBP, prolonged sitting is commonly cited in the literature, but the available evidence has not confirmed a consistent association between occupational sitting and LBP [3,4]. Nevertheless, office workers spend over two-thirds of their workday seated and this prolonged and uninterrupted sedentary behaviour [5] is a recognised public health concern [6]. The office workplace has been identified as a key setting for targeting a reduction of this ubiquitous behaviour [5]. Explicit recommendations have been provided, such as breaking up of seated-based work with standing-based work, the use of sit-stand desks or taking of short active standing breaks, to promote the avoidance of prolonged sedentary periods [7].

Prolonged standing has been associated with the development of musculoskeletal disorders including LBP for some adults [8,9]. However, causality between occupational standing and LBP could not be shown and not all people who are exposed to prolonged standing will develop LBP [10].

Previous studies [11] have used an induced pain standing task [12] to understand the characteristics that predispose a person to developing LBP during prolonged standing, but these factors are not well understood. Researchers have suggested an association between (i) reduced hip abductor muscle endurance [13,14]; (ii) reduced endurance or strength of the trunk muscles [13,15,16,17,18]; and (iii) movement control dysfunction [15,17,19] and the development of LBP during standing. However, the evidence is inconsistent and studies have used university students and community volunteers, rather than office workers [13,15,19,20,21,22].

The importance of psychosocial factors in the management of LBP is well documented in the general population [23] and in office workers [24]. However, to our knowledge, psychosocial factors have not been considered in relation to the development of standing-induced LBP in office workers. This may be because most studies investigating a prolonged standing task are conducted in young volunteers with mean age 24 ± 3 years (rather than office workers) without a history of musculoskeletal problems [13,15,16,17,18]. These inclusion criteria omit a large part of the population who have a history of LBP and do not target the population of office workers. Thus, the population of people who are often given and/or may benefit from standing workstations have been neglected from much of the previous research in this area.

The aim of this study was to identify individual, physical and psychosocial factors associated with the development (or increased severity) of LBP during a one-hour standing task. Given the findings of previous studies, we hypothesised that specific individual (history of LBP), physical (deficits in motor control and muscle endurance) and psychosocial factors (e.g., pain catastrophising and high job demands) would be associated with the development (or increased severity) of LBP during a one-hour standing task. Outcomes from this study will provide clinicians with useful information about the relationship between biopsychosocial variables and the potential development of LBP in office workers using height-adjustable workstations.

## 2. Materials and Methods

### 2.1. Participants

A convenience sample of forty office workers was recruited from the local university and surrounding community via flyers, electronic noticeboards and social media to participate in this cross-sectional laboratory study. Inclusion criteria were office-based workers who worked ≥ 30 h/week mostly sitting at a computer and were aged ≥ 18 years. Participants were excluded if they were pregnant or less than six months postpartum, had any major trauma or surgery to the spine or lower limb in the last 12 months or had a diagnosis of neurological or systemic pathology. An online survey was used to assess eligibility. The study was carried out in Brisbane (Australia), at the University of Queensland. The sample size was determined based on a power analysis, setting a probability of type I error of 0.05 (alpha), a power of 0.8 (1—probability of type II error), in order to detect a moderate effect size (Cohen’s d = 0.5) with the G*Power (v.3.1.) software [25], and the minimum sample size is *n* = 34. Based on previous work [15,21], a total of 40 participants was considered sufficient to detect differences of medium effect size, considering a significance level of 0.05 and a statistical power of 0.8. The study was conducted in accordance with the Declaration of Helsinki. The University of Queensland Human Research Ethics Committee B approved this study. This project complies with the provisions contained in the National Statement on Ethical Conduct in Human Research and complies with the regulations governing experimentation on humans (Registration: EC00457; Approval Number: #2017000666). This study was registered in the Protocol Registration and Results System (PRS) (NCT03678623).

### 2.2. Study Procedure

Participants completed a series of self-report questionnaires and attended laboratory testing conducted by a trained physiotherapist. Testing was undertaken according to a standardised protocol in the following order: (i) height and weight (ii) tests of movement control, (iii) trunk and hip muscle endurance tests, and (iv) one-hour standing task.

### 2.3. Physical Outcome Measures

One-hour standing task: the standing task consisted of participants standing for one-hour while performing their usual computer-based work [21,26]. Participants stood within a rectangular floor space (122 cm × 61 cm) with their body fist-width away from a height-adjustable workstation. The workstation was standardised to each participant so that the desk height was 5–6 cm below the lateral epicondyle, the computer monitor was at arm’s length from the body, and the top of the computer monitor was at eye level. Participants were allowed to shift their weight as often as desired but were asked to keep both feet on the ground the majority of the time. The participant was not allowed to lean on the workstation with their arms, legs or trunk. During the standing task, participants recorded the location of any pain they experienced on a body map and rated pain intensity on a 100 mm Visual Analogue Scale (VAS) at baseline, at 15 min intervals and at the end of the test. The 100 mm was anchored by “no pain” (0 mm) and “worst pain imaginable” (100 mm) [27]. The investigator verbally asked the participants to rate their pain, without access to their previous scores. The test was terminated after one hour or if the participant requested the test to be stopped or experienced pain ≥ 70 mm on the VAS.

Tests of motor control (Appendix A): the active hip abduction (AHAbd) [28], active straight leg raise (ASLR) [29] and prone knee flexion (PKF) [30] tests, which have previously established reliability, were used to assess control of movement. Each test was repeated three times on each side in random order.

In the AHAbd test, participants adopted a side-lying position and were asked to raise their top leg towards the ceiling as far as possible, keeping the knee extended, the lower limb aligned with the trunk and the pelvis aligned in the frontal plane. The difficulty in performing the test was self-rated by the participants on a 0–5 scale (ranging from 0 “no difficulty at all” to 5 “unable to lift the limb”) [28]. Quality of movement and ability to maintain the pelvis in the frontal plane were rated by the examiner on a 4-point scale (ranging from 0 “able to maintain pelvis position” to 3 “severe loss of frontal plane pelvis alignment”) [28]. The average participant-rated and examiner-rated score for each leg was used for analysis.

To perform the ASLR test, participants were positioned supine and asked to lift their leg 20 cm above the table with the knee straight and hold for 10 s. Participants rated the difficulty of performing the ASLR on a 0–5 scale (ranging from 0 “no difficulty at all” to 5 “unable to lift the limb”) [29]. The examiner scored the ASLR quality of movement by awarding one point for each of the following deviations: pelvic rotation toward the raised leg, tremor of the raised leg, slow speed of performance and verbal or nonverbal expression of difficulty by the subject. Possible scores ranged from 0 (no deviations present) to 5 (all deviations present or unable to raise the leg off the table). Scores for the left and right sides were combined to form a total score out of 10 for participant-rated and examiner-rated outcomes [29].

For the PKF, participants were positioned prone and asked to bend their knee as far as possible without moving their back [30]. The test was rated by the examiner as “correct” (a score of 0) when the knee was flexed to at least 90° without movement of the low back and pelvis, or “not correct” (a score of 1) when the low back moved into extension or rotation during knee flexion [30].

Tests of muscle endurance (Appendix A): trunk and hip muscle endurance was assessed using the following tests with published protocols and established test–retest reliability: abdominal (flexor) muscle endurance [31], side-bridge (right and left) [31], supine bridge [32], isometric hip abduction (right and left) [33] and Biering–Sorensen [31]. The tests were performed in random order, except the Biering–Sorensen, which was performed last. The objective of each test was to hold a static position for as long as possible. Each test was performed once and the time for which the position was held (seconds) was used for analysis. The test was stopped when the participant could not maintain the position (e.g., fell below horizontal for the isometric hip abduction or Biering–Sorensen tests) or elected to stop due to pain. The Biering–Sorensen test was also stopped if the maximal test time (240 s) was reached.

For the abdominal endurance test [31], participants were positioned sitting with their trunk supported in 60° of flexion, knees and hips flexed to 90° and arms crossed over the chest and feet secured. Trunk support was removed, and the participant was required to hold this position. To perform the side bridge tests [31], participants assumed a side-lying position with knees extended and the top foot in front of the bottom foot. The participant lifted their body off the ground to support their weight on their lower forearm and feet. For the supine bridge test [32], participants lay supine with knees flexed to 90°, soles of the feet on a 20 cm narrow base and hands by the ears. Participants raised their pelvis off the table and held a position with the shoulders, hips and knees in a straight line. If the participant held the position for 2 min, the dominant knee was extended and the test continued in single-leg support [32]. The isometric hip abduction test was performed with the participant in side-lying position with the pelvis secured to the plinth with straps, the bottom leg flexed at the hip and knee and a weight of 7.5% of body mass on the top ankle [33]. The participant lifted the top leg to horizontal and held this position. An inclinometer secured to the leg 10 cm superior to the lateral femoral condyle monitored any deviation from horizontal. The Biering–Sorensen test was performed in prone position. The participant’s pelvis, hips and knees were secured to a table. The trunk and upper limbs were positioned off of the table and supported on a chair [31]. To start the test, the chair was removed, and the participant was required to maintain a horizontal trunk position. Deviation from horizontal was measured using an inclinometer positioned on the mid-axillary line of the trunk at the level of the scapular.

Prior to data collection, intra-rater reproducibility for the physical tests was evaluated. The strength of agreement (measured with intraclass correlation coefficients (ICC)) was moderate (ICC: 0.4–0.6) for the AHAbd test examiner rating, ASLR examiner and participant rating and PKF test; substantial (ICC: 0.5–0.8) for the AHAbd test participant rating; and almost perfect (ICC: 0.8–1.0) for the abdominal endurance, right and left side-bridge, Biering–Sorensen test, supine-bridge and right and left isometric hip abduction test. Bland–Altman charts confirmed the results provided by the corresponding ICCs.

### 2.4. Self-Report Outcome Measures

Self-reported measures included demographics; history and severity of LBP history; total and occupational physical activity; psychosocial job characteristics; general health; and pain catastrophising.

The Nordic Musculoskeletal Questionnaire (NMQ) [34] was used to record lifetime, 12-month and 7-day prevalence of LBP. Participants were shown a body map with the low back region shaded and asked to indicate if they had experienced pain in that region (“Yes”/“No” response options) in any of the timeframes. Severity of LBP in the last 7 days was assessed with a 100 mm VAS [27].

The short form International Physical Activity Questionnaire (IPAQ) is a reliable and valid method of measuring self-report physical activity [35]. Total physical activity in MET—minutes/week—was calculated based on the sum of walking, moderate-intensity and vigorous-intensity physical activity [35]. The Occupational Sitting and Physical Activity Questionnaire (OSPAQ) was utilised to estimate time spent sitting, standing and walking at work [36]. It has acceptable reliability and validity with objective measures. Participants describe the proportion of a typical workday spent sitting and doing other physical activity. Results are expressed in minutes per day.

The abbreviated Job Content Questionnaire (JCQ) [37] was used to assess psychosocial job characteristics. It includes four domains: job control, psychological job demands, social support and physical demands. Each item was rated using a 4-point Likert. Higher scores reflect lower job control, lower psychological job demands, lower social support and lower physical demands.

The Short-Form 12 (SF-12) evaluated general health status in eight dimensions (physical function (PF), role-physical (RP), bodily pain (BP), general health (GH), vitality (VT), social function (SF), role-emotional (RE) and mental health (MH) [38], which are also reduced to Physical (PCS) and Mental (MCS) Component Summary scores. Values above or below 50 (the normative score from the general population) are interpreted as better or worse than the reference population, respectively.

The Pain Catastrophising Scale was used to assess propensity for pain catastrophising [39]. The scale contains a 5-point Likert scale (0 = “not at all” to 4 = “all the time”) on which participants rate their thoughts or feelings when experiencing pain. Scores ranged from 0 to 52 and greater scores indicate a greater degree of catastrophising.

### 2.5. Data Analysis

Participants were defined as either a pain developer (PD) or non-pain developer (NPD) based on their response to the standing paradigm protocol. PD reported a change of ≥10 mm in LBP from baseline throughout the standing paradigm. NPD were participants who did not report any change in symptoms or reported a change of <10 mm on the VAS scale throughout the standing task.

Studies investigating criteria for minimally clinically important difference (MCID) scores for pain VAS have been conducted across a range of diagnoses and populations [40]. While MCID scores are typically used to detect improvements in symptoms in response to treatment, minimal detectable change (MDC) may also be used to investigate changes in pain. Based on previous studies [11,19,20,26], the decision was made to use a change of ≥10 mm on the pain VAS at any time between start and end of the test as the cut-off point to categorise participants as PD or NPD.

### 2.6. Statistical Analysis

Statistical analyses were performed using RStudio software (R version 3.5.2, R Foundation for Statistical Computing, Vienna, Austria), with the significance level set at *p* < 0.05. Significant differences in quantitative variables between PD and NPD groups were assessed using Student’s *t*-test or Mann–Whitney’s U test (depending on the normality of the corresponding variables). The normality assumption was verified through Shapiro–Wilk’s test and graphically observed by means of kernel density plots. In the case of categorical variables, differences between sub-groups were assessed using the chi-square or Fisher’s exact test. Missing data were not considered in the computations.

Because of the small sample in some subgroups, the existence of significant and non-spurious differences was analysed using a Bayesian hypothesis test [41]. A Bayesian testing procedure proceeds in a similar way to a classical hypothesis test but, in this case, the *p*-value for adopting a decision is replaced by the named Bayes factor. This number is the ratio of probabilities of the null hypothesis and the alternative, considering the statistical information provided by the sample data. The obtained numbers were interpreted using the original Kass and Raftery’s categories [41]: Bf < 1 implies that the data are more likely under the alternative hypothesis (H1) than under the null. The strength of the likelihood can be interpreted as follows: between 1 and 0.33, anecdotal evidence; from 0.33 to 0.1, moderate evidence; from 0.1 to 0.033, strong; between 0.033 and 0.01, very strong; and <0.01, extreme.

## 3. Results

Forty office workers (22 females; mean age: 37.4 ± 6.6 years) were included in the study (Figure 1). Fourteen participants (35%) were classified as PD (due to an increase in LBP intensity of ≥10 mm during the standing paradigm), and 26 participants (65%) were NPD.

Table 1 provides the demographics, anthropometrics and self-report measures for PD and NPD groups. There were no statistically significant differences in age, sex, anthropometrics, total or occupational physical activity, psychosocial job characteristics (JCQ) or pain catastrophising between groups (all *p* > 0.12). A larger proportion of participants who developed LBP during the standing task (PD) reported experiencing LBP previously in their lifetime (64%), in the last year (93%) and in the last 7 days (79%) than those who did not experience LBP during the standing task (NPD) (all *p* < 0.05). LBP severity at the start of testing (*p* = 0.68) was similar for PD and NPD groups. The mental and physical component summary scores of the SF-12, and all dimensions except physical function (55.8 ± 2.9 vs. 49.8 ± 10, *p* = 0.01), were similar between PD and NPD participants (all *p* > 0.06).

The median holding times for the trunk and hip muscle endurance tests were lower for PD than NPD for the abdominal endurance test, Biering–Sorensen test, isometric hip abduction test (bilaterally) and the left side bridge test (all *p* ≤ 0.05; Table 2, Figure 2). Participants who were PD self-reported having greater difficulty in performing the AHAbd and ASLR tests than NPD (all *p* < 0.05; Table 2). The examiner scores for AHAbd, ASLR and PKB were not different between groups (all *p* > 0.06).

A post-hoc analysis was done to explore the effect of history of LBP in the past year on trunk and hip muscle endurance and control of movement measures (Table 3). Analyses indicate that individuals with a history of LBP have lower examiner ratings for control of movement during the ASLR (*p* = 0.01) and PKF (left side only; *p* = 0.006) tests.

## 4. Discussion

The purpose of this study was to identify demographic, physical and psychosocial factors associated with the development of LBP during a one-hour standing task in office workers. In our study, 35% of office workers developed LBP or experienced an increase in LBP intensity when standing for one hour. Our hypotheses that PD would have deficits in muscle endurance and control of lumbopelvic movement than NPD were partly supported. PD exhibited significantly reduced endurance of the trunk and hip muscles during the abdominal, side-bridge, hip abduction and trunk extensor tests, and deficits in lumbopelvic movement control, when performing an AHAbd and ASLR test compared to NPD. We correctly hypothesised that participants with a history of LBP would be more likely to develop standing-induced LBP that those without an LBP history. Our data suggest that there is a relationship between lifetime, yearly or weekly history of LBP and the development or worsening of standing-induced LBP.

Our findings of deficits in the side bridge and isometric hip abduction endurance tests, and participant-rated difficulty performing AHAbd test, in office workers who were PD suggest that decreased hip abductor endurance and/or strength may be associated with standing-induced LBP. This finding is consistent with previous studies that report decreased hip abduction [14] and side bridge [13] endurance in university students who are PD compared to those who are NPD, and differences in the AHAbd test between people who do and do not develop pain during prolonged standing [15,17,19]. Further, similar to our data, Nelson-Wong et al. [19] found that participant self-rated difficulty performing the AHAbd test was better able to predict those who developed pain during prolonged standing than the examiner-rated score. Thus, together with previous work in other populations, our data in office workers suggest that tests of hip abductor muscle function may be clinically useful for individuals who stand for prolonged periods, such office workers transitioning to a sit–stand workstation.

Our study identified that PD had poorer trunk (abdominal and lumbar) muscle strength/endurance and greater self-rated difficulty performing an ASLR than NPD. In contrast to our findings, the few studies that have assessed trunk muscle function in relation to standing-induced LBP did not find that deficits of endurance/strength [19,22] or ASLR performance [15,19] discriminated between PD and NPD. One key difference between our study and this previous work is the population studied. Our sample included individuals with and without a history of LBP, whereas other studies have included asymptomatic individuals without previous LBP. Deficits in trunk muscle function and ASLR performance have been reported in individuals with LBP [30,42]. Nelson-Wong et al. [15,43] found differences in co-activation of the trunk flexor and extensor muscles groups during a prolonged standing task in PD and NPD and suggested that trunk muscle activation may be related to standing-induced LBP. The relationship between trunk muscle strength/endurance, muscle activation patterns and standing-induced LBP requires further investigation.

Our results demonstrate that having had a prior history of LBP differentiated between PD during one-hour standing and NPD. Experiencing an episode of LBP in the last 12 months, which is a predictor of recurrence of LBP in the general population [44] and in office workers [24,45], was significantly associated with the development of pain during standing. This highlights the importance of determining LBP history in office workers who may want to transition to a sit–stand workstation. Deficits in hip [46] and trunk [42] muscle strength/endurance and control [30] have been shown to be different in individuals with and without LBP. Analyses of differences in these outcomes between study participants with and without a history of LBP in the past year identified differences in examiner-rated lumbopelvic control during ASLR and PKB tests, but no differences in muscle strength/endurance or perceived task difficulty. Thus, it is unlikely that deficits in trunk and hip muscle function in PD are purely due to a greater proportion of PD having a history of LBP.

There were no differences in most self-report variables measured between study groups, except for better physical function for NPD compared to PD. Similar to our findings, pain catastrophising was not found to be related to standing-induced LBP in individuals without a history of LBP [15,16,18]. Previous studies have found psychological characteristics predictive of LBP in an occupational setting [45]. However, the low severity of LBP in our participants, positive job control and social support and low psychological and physical demands may explain the lack of differences in psychological characteristics. It is also possible that psychological factors are more relevant in a clinical population rather than a “healthy” working population. The small sample size or ceiling effect of measures could explain our findings.

There are some limitations to this study to consider. Firstly, clinical assessments were performed by a single physical therapist. While inter-rater reliability for each measure was substantial to almost perfect, it is unknown whether the rating of movement control would be similar between different examiners. Secondly, the length of the standing task may have been insufficient to trigger differences in some outcomes. A recent systematic review suggested that clinically relevant levels of LBP were reached after 71 min of standing [8]; however, previous work by our research group and others suggests that as little as 15 min is sufficient to induce LBP [47]. Thirdly, while the sample size was based on a power analysis and on previous studies, it was small for some subgroups of variables and only medium effect sizes could be determined. Bayesian analysis was performed to address this concern.

Findings from this study suggest that a history of LBP and deficits in hip abductor and trunk muscle strength/endurance and participant-rated difficulty performing tasks that challenge lumbopelvic control are associated with the development of standing-induced LBP in office workers. Clinicians could consider assessing hip and trunk muscle function in office workers who are planning to transition to a sit–stand workstation. It is important to note that office workers in this study stood for one hour without sitting, resulting in 35% of participants developing standing-induced LBP. It is recommended that office workers seeking advice regarding the use of height adjustable workstations be advised to gradually upgrade time spent standing to enable the hip and trunk muscles to adjust to increased demands during standing [48] and change position regularly [11,49].

## 5. Conclusions

Office workers who reported LBP during a one-hour standing task in this cross-sectional study presented with decreased hip and trunk muscle endurance, self-reported difficulty with tasks that challenge lumbopelvic control and an increased likelihood of having a history of LBP than NPD. Findings suggest that these outcomes may be important to evaluate in office workers who are transitioning to standing workstations. Prospective research is needed to determine if these measures are able to identify office workers who develop standing LBP when transitioning to a sit–stand workstation. While this was not a treatment study, and data were collected at one point in time, the researchers propose that addressing hip and trunk muscle deficits may help to reduce the risk of developing LBP when using a standing workstation. Future research should look at the effect of addressing hip and trunk muscle deficits on standing induced LPB in office workers.

## Figures and Tables

**Figure 1 ijerph-17-07104-f001:**
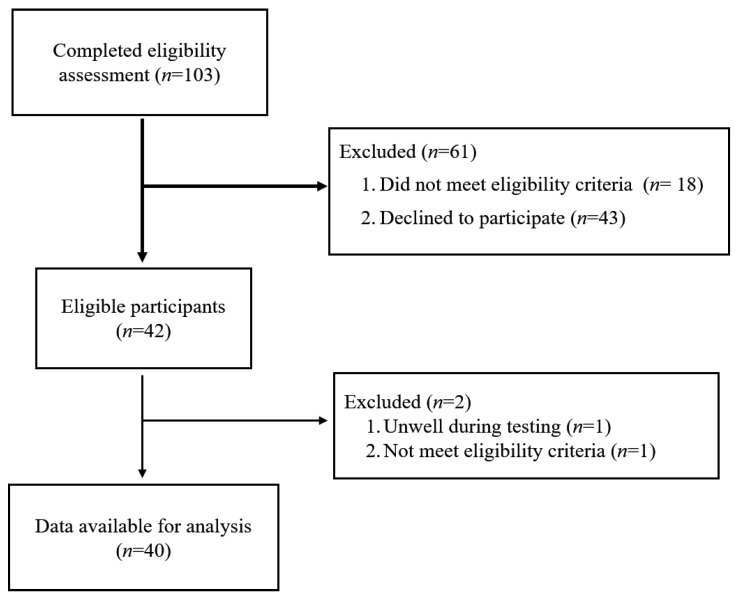
Flow diagram.

**Figure 2 ijerph-17-07104-f002:**
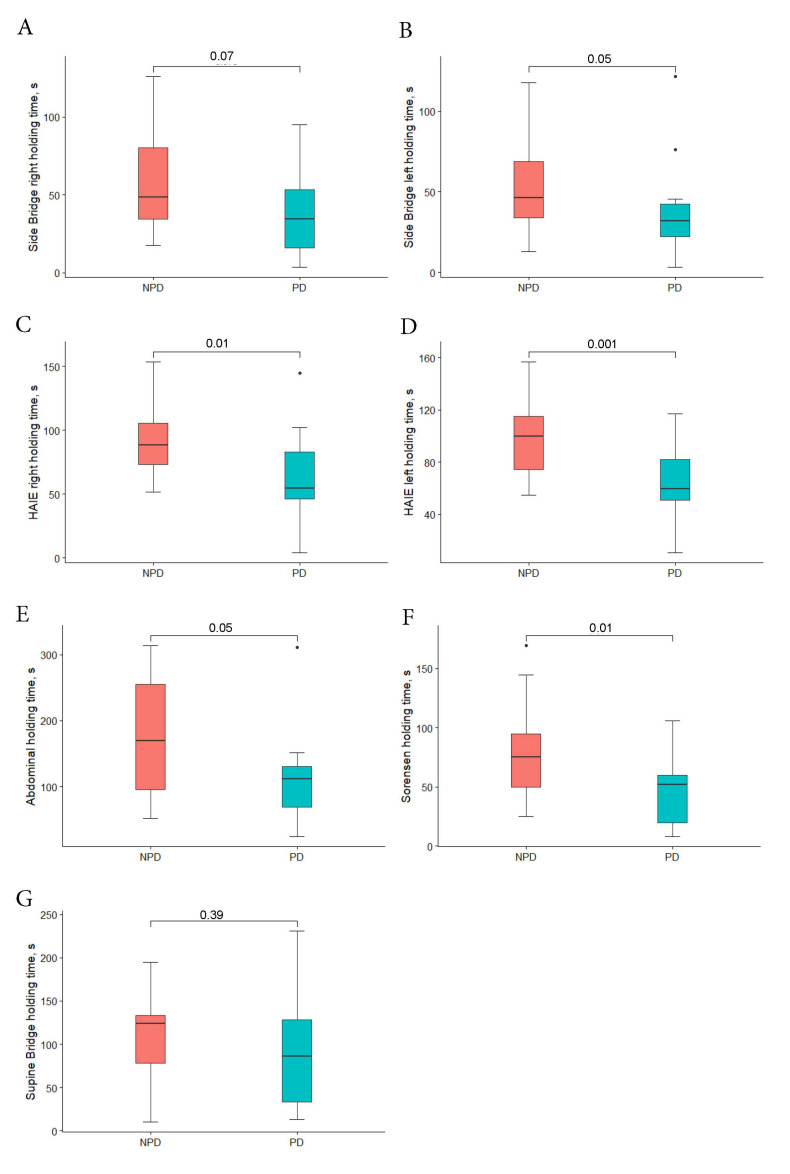
Endurance test. (**A**) Side bridge right side, (**B**) side bridge left side, (**C**) HAIE right side, (**D**) HAIE left side, (**E**) abdominal, (**F**) Sorensen, (**G**) supine bridge holding times for individual NPD (*n* = 26) and PD (*n* = 14). In the box plot, red boxes represented NPD and blue boxes represent PD; horizontal bold lines show the medians, and the box limits indicate the 25th and 75th percentiles. Whiskers extend one and a half times the length of the box (interquartile range), and dots represent the outliers. *p* obtained from the Mann–Whitney U test or t-Student test for independent samples. NPD, non-pain developers; PD, pain developers; s, seconds; HAIE, Hip Abductor Isometric Endurance test.

**Table 1 ijerph-17-07104-t001:** Differences in demographics and self-report measures for non-pain developers (NPD) and pain developers (PD).

	NPD (*n* = 26)	PD (*n* = 14)		
			*p*	Bf
Age (years), mean ± SD	36.3 ± 8.6	39.4 ±11.5	0.51	2.2
Sex (female), *n* (%)	13 (50.0)	9 (64.3)	0.39	
BMI (kg/m^2^), mean ± SD	26.2 ± 4.3	26.5 ± 7.8	0.49	3.1
IPAQ, MET—min/week (median)	2138.5 ± 1897.5	2357.0 ± 978	0.80	2.7
History of LBP				
LBP Lifetime			0.04	0.35
No (*n*, %)	18 (69.2)	5 (35.7)		
Yes (*n*, %)	8 (30.8)	9 (64.3)		
LBP last year			0.03	0.31
No (*n*, %)	10 (38.5)	1 (7.1)		
Yes (*n*, %)	16 (61.5)	13 (92.9)		
LBP last 7 days			0.02	0.14
No (*n*, %)	16 (61.5)	3 (21.4)		
Yes (*n*, %)	10 (38.5)	11 (78.6)		
LBP min 0			0.08	-
No (*n*, %)	24 (92.3)	10 (71.4)		
Yes (*n*, %)	2 (7.7)	4 (28.6)		
LBP severity, last 7 days (0–100), mean ± SD	30.4 (31.2)	51.2 (21.8)	0.08	0.86
LBP severity, start of testing (0–100), mean ± SD	12.7 (3.7)	10.0 (7.5)	0.68	-
OSPAQ, mean ± SD				
Sitting per week (min)	2126.1 ± 517.6	1929.9 ± 175.8	0.09	1.50
Sitting per workday (min)	430.6 ± 102.8	391.9 ± 85	0.16	1.80
Standing per week (min)	281 ± 250.2	221.2 ± 190.8	0.64	2.50
Standing per workday (min)	57.1 ± 50.3	40.5 ± 30	0.54	1.9
Walking per week (min)	179.3 ± 144.2	210.4 ± 281.2	0.68	2.9
Walking per workday (min)	36.3 ± 28.6	39.7 ± 43.1	0.67	3.0
PCS, mean ± SD				
Rumination	3.5 ±3.4	5.3 ±4.2	0.18	1.32
Magnification	1.7 ±1.3	2.9 ±2.9	0.29	0.84
Helplessness	2.7 ±3.7	4.6 ±4.8	0.16	1.51
PCS total (0–52)	7.8 ±7.2	12.8 ±10.2	0.12	0.93
JCQ, mean ± SD				
Job Control (24–96)	46.7 ± 13.2	42.4 ± 10.5	0.31	2.0
Psychological Job Demands (3–12)	8 ± 1.7	7.6 ± 1.4	0.34	2.5
Social Support (4–16)	7.5 ± 1.7	7.1 ± 1.8	0.62	2.8
Physical Demands (2–8)	7.5 ± 0.8	7.6 ± 0.9	0.40	2.9
SF-12, Normal-Based Scores (50 ± 10)				
Vitality	52.1 ± 9.1	54.0 ± 9.3	0.29	2.68
Social Function	51.1 ± 7.9	49.3 ± 9.1	0.44	2.64
Role Emotional	43.9 ± 12.5	47.0 ± 11.7	0.34	2.48
Mental Health	48.1 ± 9.6	53.1 ± 6.5	0.09	0.96
Mental Component Summary	45.6 ± 9.7	51.4 ± 8.7	0.06	0.83
Physical Function	55.8 ± 2.9	49.8 ± 10	0.01	0.13
Role Physical	49.8 ± 9.3	49.3 ± 12.9	0.66	3.10
Bodily Pain	52.5 ± 5.8	47.4 ± 10.5	0.12	0.69
General Health	52.2 ± 8.8	49.9 ± 12.1	0.72	2.59
Physical Component Summary	55.4 ± 5.0	48.8 ± 10.6	0.09	0.21

*p*-values indicate statistically significant differences in having had a prior history of LBP (lifetime, last year, last 7 days) and physical function between groups. Bf, Bayesian factor; BMI, body mass index; IPAQ, International Physical Activity Questionnaire; MET, Metabolic Equivalent of Task (computed as the sum of walking, moderate-intensity and vigorous-intensity physical activity); LBP, low back pain; OSPAQ, Occupational Sitting and Physical Activity Questionnaire; min, minutes; PCS, Pain Catastrophising Scale; JCQ; Job Content Questionnaire.

**Table 2 ijerph-17-07104-t002:** Differences in physical outcome measures for non-pain developers (NPD) and pain developers (PD).

	NPDs (*n* = 26)	PDs (*n* = 14)		
			*p*	Bf
Tests of trunk and hip muscle endurance				
Side bridge right side (s)	57.3 ± 28.9	39.4 ± 29.9	0.07	0.84
Side bridge left side (s)	52.6 ± 25.6	38.1 ± 30.1	0.05	1.14
Isometric hip abduction (right leg) (s)	90.7 ± 25.6	62.4 ± 36.5	0.01	0.15
Isometric hip abduction (left leg) (s)	97.8 ± 27.9	62.2 ± 32.2	0.001	0.02
Supine bridge (s)	101.1 ± 51	89.2 ± 61.3	0.39	2.64
Abdominal (s)	172.1 ± 86.4	112.0 ± 69.6	0.05	0.47
Sorensen (s)	78.6 ± 36.3	48.7 ± 31.9	0.01	0.25
Tests of movement control				
AHAbd, right side, examiner score (0–3)	1.4 ± 0.7	1.6 ± 1	0.69	2.74
AHAbd, left side, examiner score (0–3)	1.6 ± 0.6	1.8 ± 0.9	0.33	1.99
AHAbd, right side, participant score (0–5)	1.5 ± 0.9	2.2 ± 0.9	0.04	0.56
AHAbd, left side, participant score (0–5)	1.5 ± 0.9	2.4 ± 1.1	0.03	0.21
ASLR, total examiner score (0–10)	2.5 ± 1	2.8 ± 1.1	0.42	2.38
ASLR, total participant score (0–10)	2.2 ± 1.4	3.4 ± 1.7	0.03	0.39
PKF, right side, examiner score (0–1)	0.7 ± 0.4	0.6 ± 0.4	0.39	2.30
PKF, left side, examiner score (0–1)	0.4 ± 0.4	0.6 ± 0.4	0.06	0.81

Data are presented as mean ± SD. *p*-values indicate statistically significant differences in trunk and hip muscle endurance (side bridge left side, isometric hip abduction, abdominal and Sorensen test) and in movement control (AHAbd and ASLR participant score). Bf, Bayesian factor; s, seconds; AHAbd, active hip abduction; ASLR, active straight leg raise; PKF, prone knee flexion (0 = correct; 1 = not correct).

**Table 3 ijerph-17-07104-t003:** Differences in physical outcome measures for participants with and without history of lower back pain (LBP) in the past year.

	LBP—Last Year (*n* = 29)	No LBP—Last Year (*n* = 11)	*p*
Test of trunk and hip muscle endurance			
Side bridge right side (s)	48.9 ± 29.9	56.7 ± 31.6	0.52
Side bridge left side (s)	44.0 ± 26.0	56.9 ± 31.2	0.31
Isometric hip abduction (right leg) (s)	78.5 ± 36.2	86.8 ± 19.6	0.67
Isometric hip abduction (left leg) (s)	81.1 ± 35.3	96.5 ± 27.7	0.06
Supine bridge (s)	88.1 ± 57.4	120.3 ± 38.3	0.09
Abdominal (s)	143.2 ± 78.9	171.8 ± 100.6	0.44
Sorensen (s)	66.4 ± 36.7	72.4 ± 40.4	0.52
Test of movement control			
AHAbd, right side, examiner score (0–3)	1.5 ± 0.8	1.2 ± 0.7	0.14
AHAbd, left side, examiner score (0–3)	1.7 ± 0.7	1.5 ± 0.7	0.32
AHAbd, right side, participant score (0–5)	1.9 ± 0.9	1.3 ± 0.8	0.10
AHAbd, left side, participant score (0–5)	1.9 ± 1.1	1.4 ± 0.8	0.15
ASLR, total examiner score (0–10)	2.9 ± 0.9	1.9 ± 1.0	0.01
ASLR, total participant score (0–10)	2.6 ± 1.6	2.3 ± 1.5	0.99
PKF, right side, examiner score (0–1)	0.7 ± 0.4	0.5 ± 0.5	0.12
PKF, left side, examiner score (0–1)	0.6 ± 0.4	0.2 ± 0.3	0.006

Data are presented as mean ± SD. s, seconds; AHAbd, active hip abduction; ASLR, active straight leg raise; PKF, prone knee flexion.

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
