# Peer review of "What Psychosocial and Physical Characteristics Differentiate Office Workers Who Develop Standing-Induced Low Back Pain? A Cross-Sectional Study"

_ijerph, 2020, doi:10.3390/ijerph17197104_

Round 1
Reviewer 1 Report
The manuscript reports on what psychosocial and physical characteristics differentiate office workers who develop standing induced low back pain.
I have some concern:
- A cross-sectional survey with 40 participants is not enough to get a true conclusion.
- The Introduction does not provide sufficient background on LBP.
“It is recognized that prolonged standing is associated with the development of LBP” – it is not. Not sitting or standing is leading to LBP. It is a myth – see some references, you can use also the back pain series from LANCET 2018:
- “Sit Up Straight”: Time to Re-evaluate. J Orthop Sports Phys Ther 2019;49(8):562–564.
- Caneiro JP, Bunzli S, O'Sullivan P. Beliefs about the body and pain: the critical role in musculoskeletal pain management. Brazilian Journal of Physical Therapy. 2020 Jun 20.
- Lin I, Wiles L, Waller R, Goucke R, Nagree Y, Gibberd M, Straker L, Maher CG, O'Sullivan PPB. What does best practice care for musculoskeletal pain look like? Eleven consistent recommendations from high-quality clinical practice guidelines: systematic review. Br J Sports Med. 2019
- Olugbade, et al. The relationship between guarding, pain, and emotion. PAIN 2019
I wonder why you cozen Prolonged standing task? “ we hypothesized that participants who develop LBP during standing would have (i) decreased lumbopelvic and hip muscle endurance and control” you hypothesis that this is the reason for LBP? We know by now that it isn’t the case.
“Findings from this study suggest that a history of LBP and deficits in hip abductor and trunk
muscle strength/endurance and participant-rated difficulty performing tasks that challenge
lumbopelvic control are associated with the development of standing-induced LBP in office workers. Clinicians could consider assessing hip and trunk muscle function in office workers who are planning
to transition to a sit-stand workstation.”
I can agree with the history of LBP – we know it by now from research, but do you think that office worker who has deficits in hip abductor and trunk muscle strength/endurance and participant-rated difficulty performing tasks that challenge lumbopelvic control – will have more pain? and if we will teach them this exercise they will be better? We know that physical activity is the best treatment for LBP – but no specific exercise is better – so your conclusion is problematic.
“Further research is needed to understand the role of hip and trunk strengthening exercises in preparing office workers to transition to a sit-stand workstation and in preventing the development of standing-induced LBP”
We don’t need more research in this field, we need to implement what we know already (see LANCT series “call for action”)
Author Response
Please see the attachement.

Reviewer 2 Report
Greetings: The authors have presented solids data that someday may make valuable contributions to modern-world tech and office workers. The implications for designing interventions to reduce back pain are clearly outlined.
Some details are unclear which makes it difficult for non-experts to appreciate the findings, and even for some experts to wonder how to replicate the findings, though these are minor and easily corrected. I note those next:
Lines 53, 70, and 219 could be coordinated so that the reader has the same age data (range and average) in each location. That allows for immediate appreciation of why your work with older office workers is additionally important.
Lines 121 through 152 are difficult to visualize and unless the reader is already familiar with the tests, it is doubtful the results could be easily replicated, if at all. Either consider a few simple figures of bodies -- perhaps stick figures -- depicting the positions and movements, or explain the positions and instructions more clearly. I have no idea how to translate the instructions in lines 121, 123, and 124; I do not know what "high sitting" means in line 134; I suspect you mean 90 degrees instead of 900 in line 140; and I cannot visualize the remaining lines through 152.
In line 155 the authors write that a .40 coefficient is of "moderate" strength even though this fails to account for around 85% of the variance. This may be statistically acceptable but any measure with a reliability of .40 is of virtually no practical value.
I find the conclusions carefully written, and the recommendations in lines 335-339 vey useful.
In line 348 I wonder if the word "address" should be eliminated.
Except for these minor comments, I find this work well done!
Reviewer 3 Report
1. There are only 40 subjects, and it would be nice if the study was conducted with more subjects.
2. All of the evaluation tools used in the research method were subjectively reported by the subject, but an additional use of objective evaluation tools that can derive objective results is required.
3. The Figure 2 is not in good condition, so it would be better to express it in another way.
4. Check the references. Read the instruction to authors in detail.
Reviewer 4 Report
What psychosocial and physical characteristics differentiate office workers who develop standing induced low back pain? A cross-sectional study
The article is interesting, but could improve with some modifications.
- Summary: The design is transversal, so you must modify examine by describing
- Introduction: line 50-51: It is a statement that must be based on the literature and not on your knowledge and experience.
- You do not need to make a hypothesis, as it is not consistent with the epidemiological design.
- Material and methods
The objectives must be entered in this section
The selected design is not consistent with the objectives and results. In a cross-sectional study, causal relationships or comparisons between samples cannot be established:
In medical research, social science, and biology, a cross-sectional study (also known as a cross-sectional analysis, transverse study, prevalence study) is a type of observational study that analyzes data from a population, or a representative subset, at a specific point in time—that is, cross-sectional data.
Specify where the research was carried out (place, province, university). Justify why a convenience and non-representative and random sampling is selected if you want to establish causal relationships. Justify why n was chosen (references 20 and 21 are not enough to justify such a small n if you want to extrapolate results).
- Podría añadir el dictamen del comité de ética, dado el carácter experimental del estudio. (I could add the opinion of the ethics committee, given the experimental nature of the study.)
Reviewer 5 Report
This study investigated the demographic, physical and psychosocial factors associated with an increase in LBP during a one-hour standing task. Forty office workers in the local university and community were recruited and responded a series of surveys and questionnaires, as well as performed a one-hour standing test. The participants were thus divided into two groups, namely pain developers (PD) and non-PD (NPD), based on the pain VAS result. In the manuscript, there are many deficits and unclear descriptions need to be clarified or improved in the future study.
1. One of my most concerns was how the authors determined the cut point between PD and NPD groups. As stated by the authors, based on previous studies, a relatively low level pain-inducing stimulus was used in this study, the decision was made to use a change of ≥10 mm on the pain VAS at any time between start and end of the test as the cut-point to categorize participants as PD or NPD. This point is extremely important because if the cut-point changes, the study result may differ. Unfortunately, the statement “relatively low level” is very vague and non-scientific. Is any evidence or reference for this pain threshold identification? In my opinion, a change in pain of more than or equal to 10 mm in a whole 100-mm scale is easy to be observed during a 1-h standing period, especially in a subjective rating. This may cause the results biased regarding both miss and false alarm between the PD and NPD groups, and the results may be misled. In addition, the results may also be influenced by how the participants’ standing posture was requested during the test; the more posture controlled, the more pains occurred. The standing posture that the participants were requested may be not similar to that their works as usual. I doubt if this test can be repeated by other studies and the similar results could be achieved.
2. In the study, 40 office workers in the local university and community were recruited. This sample size is too small even though the Bayes Factors (BF) were used. However, the significant terms among the responses almost showed a moderate evidence, that is, BF ranged from 1/3 to 0.1.
3. The term “prolonged standing” was used in the manuscript. In general, a standing period more than 2-hrs or even 4-hrs was defined as the prolonged standing as performed in the previous studies. There is only 1-hr adopted in this test.
4. In Introduction, the section should be more clearly stated and, in my opinion, the Introduction section should initially present a more general approach and gradually address the problem (gap) and then present the objective.
5. Some references cited in the texts should be checked for their appropriateness and completeness throughout the manuscript. For example, L38 [3,4], these two papers cannot support the corresponding statement completely.
6. L153, intra-rater reproducibility, I am not sure if this was for intra-subject reproducibility, not for the rater.
7. Figures 1 and 2 are too small and unclear to be read.
Round 2
Reviewer 1 Report
I review the revised version.
The manuscript has been improved, but the research design and the results still don't support the conclusion.
Kind regards,
Author Response
"Please see the attachment."

Reviewer 4 Report
Once the manuscript has been reviewed, it is requested to answer the following questions:
1. OK
2. OK
3. Author coment:
We agree that a hypothesis is not needed in cross-sectional studies, but we have followed the STROBE Statement (items that should be included in reports of cross-sectional studies) which recommends, “State specific objectives, including any prespecified hypotheses” (Objective 3).
Response:
Detail in the manuscript following the objectives that is a prespecified hypotheses
4 Author coment:
We would like to clarify that the aim of this study was to identify individual, physical, and psychosocial factors associated with the development (or increased severity) of LBP during a one-hour standing task. The results are not investigating causal relationships nor comparisons between two groups. Our study describes two groups and explores the association between variables considering individual, physical and psychosocial factors as possible predictive factors and pain severity during a one-hour standing task as a dependent variable.
Response:
The cross-sectional designs collect the study variables simultaneously and their analysis unit is individual people. They are useful in determining the prevalence and allow to establish associations between variables quickly. Being a cross-sectional study, this type of temporality does not ensure that the exposure has preceded the outcome because there is no follow-up over time. the risk or prediction could not be directly determined, as this is reserved for studies with a longitudinal temporal focus
5. Author coment: (I could add the opinion of the ethics committee, given the experimental nature of the study.)
The University of Queensland Human Research Ethics Committee B approved this study: This project complies with the provisions contained in the National Statement on Ethical Conduct in Human Research and complies with the regulations governing experimentation on humans”. The name of Ethics Committee representative is Dr. Frederick Khafagi. Registration: EC00457. Approval Number: 2017000666.
Response: It´s important including this information in the manuscript.
Author Response
"Please see the attachment."

Reviewer 5 Report
I have to say that the authors definitely do a very good job for modifying their manuscript. Many previous studies are provided to support the idea for identifying the PD and NPD, as well as a smaller sample size, even though I still doubt the appropriateness of the method. However, the protocol is now rational. Other issues what I concerned have also been well addressed in the revision. Only one problem is the low quality of Figure 1. The figure is blurred. This issue is minor and may be corrected in the proofreading stage.
Author Response
"Please see the attachment."
